# Modelling Peroxisomal Disorders in Zebrafish

**DOI:** 10.3390/cells14020147

**Published:** 2025-01-20

**Authors:** Chenxing S. Jiang, Michael Schrader

**Affiliations:** Biosciences, Faculty of Health and Life Sciences, University of Exeter, Exeter EX4 4QD, UK; cj501@exeter.ac.uk

**Keywords:** peroxisomes, lipid metabolism, PEX, organelle biogenesis, metabolic disorders, fatty acid beta-oxidation, FIS1, VWA8, *Danio rerio*

## Abstract

Peroxisomes are ubiquitous, dynamic, oxidative organelles with key functions in cellular lipid metabolism and redox homeostasis. They have been linked to healthy ageing, neurodegeneration, cancer, the combat of pathogens and viruses, and infection and immune responses. Their biogenesis relies on several peroxins (encoded by *PEX* genes), which mediate matrix protein import, membrane assembly, and peroxisome multiplication. Defects in peroxins or peroxisomal enzymes can result in severe disorders, including developmental and neurological abnormalities. The drive to understand the role of peroxisomes in human health and disease, as well as their functions in tissues and organs or during development, has led to the establishment of vertebrate models. The zebrafish (*Danio rerio*) has become an attractive vertebrate model organism to investigate peroxisomal functions. Here, we provide an overview of the visualisation of peroxisomes in zebrafish, as well as the peroxisomal metabolic functions and peroxisomal protein inventory in comparison to human peroxisomes. We then present zebrafish models which have been established to investigate peroxisomal disorders. These include model zebrafish for peroxisome biogenesis disorders/Zellweger Spectrum disorders, and single enzyme deficiencies, particularly adrenoleukodystrophy and fatty acid beta-oxidation abnormalities. Finally, we highlight zebrafish models for deficiencies of dually targeted peroxisomal/mitochondrial proteins. Advantages for the investigation of peroxisomes during development and approaches to the application of zebrafish models for drug screening are discussed.

## 1. Introduction

Peroxisomes are single membrane-bound, oxidative organelles defined by a fine, granular matrix that are ubiquitously found in all eukaryotes (some exceptions exist). Since their discovery 70 years ago [1], the so-called “microbodies” have been associated with key metabolic and non-metabolic cellular functions. Nobel laureate Christian De Duve was the first to isolate and characterise the organelle functionally, revealing its oxidative nature, reflected by the identification of several H_2_O_2_-generating flavin-oxidases and catalase, one of its prominent marker enzymes, which decomposes H_2_O_2_ into water and oxygen. Based on these findings, De Duve suggested the name “peroxisome” for the organelle. This functional term subsequently replaced the morphological term “microbody” [2]. The importance of catalase and other peroxisomal enzymes in protecting the cell from oxidative damage by reactive oxygen species (ROS) has linked peroxisomes to the regulation of cellular redox balance, cellular ageing, age-related disorders, and cancer [3,4,5]. It should be noted that H_2_O_2_ is also an important signalling molecule [6]. In line with this, peroxisomal contributions to cellular signalling, the combat of pathogens and antiviral defence have been revealed [7,8,9,10]. More recently, additional roles of peroxisomes in infection and immune responses have been described [11].

Aside from their role in cellular redox homeostasis, peroxisomes have important functions in cellular lipid metabolism [12]. In mammals, these include a peroxisomal beta- and alpha-oxidation pathway for the degradation of fatty acids, the biosynthesis of ether phospholipids (e.g., myelin sheath lipids), bile acids, and polyunsaturated fatty acids (PUFAs), e.g., docosahexaenoic acid (DHA), which fulfils important roles in the brain and retina [13]. Mammalian peroxisomes also contribute to other pathways not related to lipid metabolism such as purine, polyamine, glyoxylate, and amino acid metabolism [12]. Whereas in plants and yeast only a peroxisomal beta-oxidation pathway exists, animals possess both a peroxisomal and a mitochondrial beta-oxidation pathway, both of which include organelle-specific enzymes. A key enzyme in peroxisomal fatty acid beta-oxidation is acyl-CoA oxidase (ACOX1), which mediates the first step of the pathway, generating H_2_O_2_ (mitochondria contain an acyl-CoA dehydrogenase instead) (Figure 1). Peroxisomal beta-oxidation shows substrate specificity for more complex fatty acids, such as very long chain fatty acids (VLCFA), bile acid intermediates, long chain dicarboxylic acids (DCAs), eicosanoids, and the side chains of certain xenobiotics that cannot be degraded in mitochondria. However, unlike mitochondria, peroxisomes cannot oxidise fatty acids to completion, and only chain-shorten fatty acids, which are routed to mitochondria for further oxidation [14,15] (Figure 1). Peroxisomes also contain enzymes for the alpha-oxidation of branched chain fatty acids, such as phytanic acid, which humans receive through dairy products or ruminant animal fats. In addition, peroxisomes are essential to the creation of the ether-bond during the synthesis of ether phospholipids and plasmalogens. This biosynthetic pathway starts in the peroxisomes and is completed in the endoplasmic reticulum (ER) [12,16]. The intimate metabolic cooperation between peroxisomes and the ER in ether lipid biosynthesis or between peroxisomes and mitochondria in fatty acid beta-oxidation illustrates that peroxisomes do not function in isolation. They cooperate with several other subcellular organelles including the ER, mitochondria, lipid droplets, and lysosomes [17]. This cooperation is supported by the formation of membrane contact sites, e.g., with the ER involving the peroxisomal tether protein ACBD5 and ER-resident VAP proteins [18,19] (Figure 2).

Peroxisomes are dynamic organelles with high plasticity. They respond to environmental changes with astonishing changes in their numbers, morphology (e.g., size, shape), and enzyme composition. Their biogenesis depends on a number of peroxins (encoded by *PEX* genes), which are still being discovered. Peroxins are peroxisomal biogenesis factors, which are essential for matrix protein import, membrane assembly, and multiplication/proliferation (reviewed in [20]) (Figure 2).

**Figure 2 cells-14-00147-f002:**
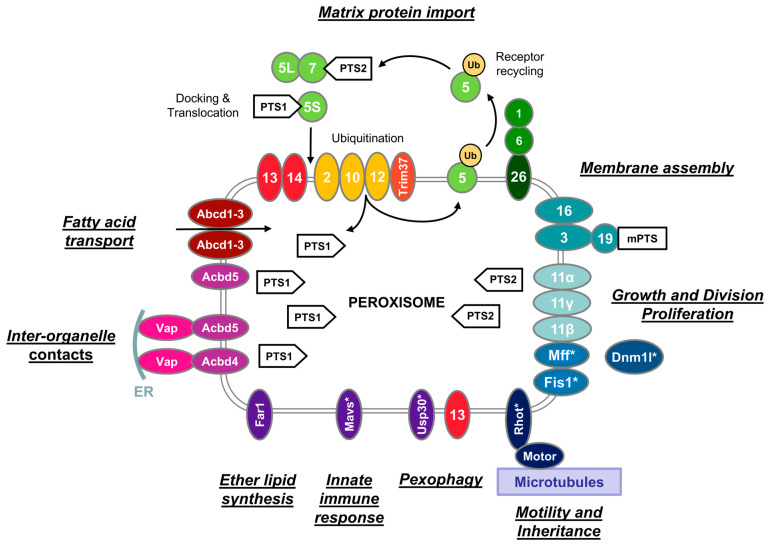
Schematic overview of predicted functions of Pex proteins and peroxisomal membrane proteins in zebrafish. **Matrix protein import**: Following translation on free ribosomes, cargo proteins that contain the peroxisomal targeting signals (PTS1 or PTS2) bind their respective cytosolic receptors, Pex5 or Pex7 (Pex proteins are labelled as numbers in the figure). For PTS2 import, accessory factors like the long isoform of Pex5 (Pex5L) are required. The receptor-cargo complexes dock at the peroxisomal membrane via Pex13 and Pex14, followed by translocation into the matrix through a dynamic translocon, Pex2, Pex10, and Pex12. Here, a pore-like structure of Pex13/Pex14 is suggested, which allows the import of completely folded, co-factor bound proteins. Export of the receptor back to the cytosol involves ubiquitination (Ub) and extraction by an AAA–ATPase complex, Pex1, and Pex6, with Pex6 interacting with Pex26. **Membrane assembly and PMP insertion**: The integration of peroxisomal membrane proteins (PMPs) with targeting signals (mPTS) is mediated by Pex19, Pex3, and Pex16. Pex19 acts as a cycling receptor/chaperone, binding PMPs in the cytosol and interacting with Pex3 at the peroxisomal membrane. **Growth and division, proliferation**: Pex11α, Pex11β, and Pex11γ regulate peroxisome size and number. Pex11β remodels the peroxisomal membrane and recruits the fission machinery, including the membrane adaptors Mff and Fis1, which engage with the dynamin-like GTPase Dnm1l to drive fission. **Motility and inheritance**: In mammals, peroxisomes move along microtubules via Rhot/Miro proteins, which serve as membrane adaptors for microtubule-dependent motor proteins. **Inter-organelle contacts**: Peroxisome-ER membrane contact sites are mediated by Acbd5 and Acbd4, which bind ER-resident Vap proteins. **Fatty acid transport**: Fatty acids enter peroxisomes through ABC transporters (Abcd 1-3), and ACBD5 captures VLCFAs for transport in mammals. **Other peroxisomal membrane proteins**: Zebrafish peroxisomal membrane proteins include Far1 (fatty acyl-CoA reductase 1, involved in ether lipid biosynthesis), Mavs (mitochondrial antiviral signalling protein, involved in innate immune response), Usp30 (ubiquitin-specific protease 30, a deubiquitinase involved in the turnover/pexophagy of peroxisomes), and Trim37 (tripartite motif-containing protein 37, an E3 ubiquitin-protein ligase that aids in Pex5-mediated protein import). Some proteins may localise to both peroxisomes and mitochondria (marked with asterisks). Abbreviations: Pex, peroxin; PMP, peroxisomal membrane protein; mPTS, membrane protein targeting signal (adapted from [21]).

Mutations in *PEX* genes cause severe peroxisome biogenesis disorders (PBDs), e.g., Zellweger spectrum disorders [22]. These severe disorders are characterised by a loss of peroxisomal integrity and metabolic functions, resulting in developmental and neurological abnormalities. Besides PBDs, several single enzyme deficiencies (SEDs) have been described (e.g., adrenoleukodystrophy), which are caused by mutations in individual peroxisomal enzymes/proteins and (in contrast to PBDs) usually affect only one peroxisomal metabolic pathway [13,23]. The lack of functional peroxisomes in PBDs or enzyme deficiencies in SEDs result in an accumulation of peroxisomal substrates (e.g., VLCFA, bile acid intermediates, phytanic acid), which can no longer be degraded and are toxic to the cell and organism. Furthermore, physiologically essential molecules synthesised by peroxisomes (e.g., ether lipids/plasmalogens, docosahexaenoic acid, bile acids) become deficient. 

Their essential roles in human health and development and involvement in cellular/organismal lipid metabolism, redox homeostasis, healthy ageing, neurodegeneration, cancer, the combat of pathogens and viruses, and infection and immune responses have sparked great interest in peroxisomes and their biology. The need to understand peroxisomal functions in tissues and organs, including during development, has led to the establishment of vertebrate models. Several mouse models have been developed to study the loss of peroxisome function (summarised in [24]). However, the zebrafish (*Danio rerio*) has also become an attractive vertebrate model organism to investigate peroxisomal functions. Zebrafish are particularly suited for studying lipid metabolism in the context of lipid-related diseases [25,26]. Furthermore, their strong genetic similarity to mammals makes them an excellent model for researching developmental and neurological processes. Advantages of zebrafish include rapid development, short generation time, low maintenance costs, ease of genetic manipulation, and the optical transparency of embryos. Combined with advanced imaging techniques, these features enable in vivo visualisation of biological processes at the organismal level [21,27]. In addition, they are highly suitable for drug screening approaches [28]. 

In the following, we will provide an overview of the visualisation of peroxisomes in zebrafish as well as the peroxisomal metabolic functions and peroxisomal protein inventory. We will then present current zebrafish models which have been established to investigate peroxisomal disorders.

## 2. Visualisation of Peroxisomes in Zebrafish 

Peroxisomes were initially visualised in zebrafish embryos and adults using diaminobenzidine (DAB)-cytochemistry [29,30]. The DAB method depends on the peroxidatic activity of peroxisomal catalase, and it led to the identification of peroxisomes as a ubiquitous subcellular organelle in eukaryotes [31]. It results in an electron-dense precipitate in peroxisomes, which can be visualised by light and electron microscopy. In zebrafish, peroxisomes were found to be abundant in the liver, renal proximal tubules, and intestinal epithelium, which is similar to mammals including humans. Upon the exposure of zebrafish to potent peroxisome proliferators such as fibrates or phthalate esters, hepatic peroxisomes respond with an increase in number [32,33,34]. This is similar to observations in rodents and other aquatic organisms, where peroxisome proliferation has been employed as an indicator for environmental pollution [35]. Peroxisome proliferators act via Peroxisome Proliferator-Activated Receptors (PPARs), nuclear hormone receptors that regulate genes associated with peroxisome biogenesis and lipid metabolism. These receptors have also been identified in zebrafish [36].

With the discovery of green/red fluorescent proteins (GFP, RFP) and peroxisomal targeting signals (PTS), GFP/RFP-PTS1 fusion proteins became available, which when expressed, label peroxisomes in zebrafish embryos, fish cells, or adult fish [37,38]. PTS1 signals are C-terminal sequences composed of about 12 amino acids. However, the addition of the canonical tripeptide SKL at the very C-terminus of GFP/RFP is sufficient to target these proteins to peroxisomes in many organisms, including zebrafish. A recent study analysing peroxisomal targeting sequences in zebrafish found that, similar to other vertebrates, most peroxisomal matrix proteins in *D. rerio* possess a PTS1, while only a small subset contain a PTS2 (an N-terminal nonapeptide) [21]. While the overall characteristics of PTS1 motifs are comparable between zebrafish and humans, variations were noted at the level of specific individual proteins.

Probes for the in vivo labelling of peroxisomes have been scarce [39]. Very recently, novel fluorescent fatty acid conjugates were developed for live-cell imaging of peroxisomes [40]. The PeroxiSPY650 (far red) and PeroxiSPY555 (red) probes offer high specificity for peroxisomes, bright fluorescence, and rapid, non-toxic staining, which makes them well-suited for live-cell, animal, and deep tissue imaging of peroxisomes. In zebrafish embryos, the PeroxiSPY dye successfully stained peroxisomes, although longer incubation times were necessary for penetration into deeper tissue levels [40].

## 3. Peroxisomal Protein Inventory and Metabolic Pathways in Zebrafish

A recent study provided a detailed analysis of the peroxisomal protein repertoire in zebrafish and its associated metabolic pathways [21]. *D. rerio* encodes orthologues of key human proteins responsible for peroxisome biogenesis, dynamics, and metabolic functions, such as fatty acid oxidation, ether lipid biosynthesis, purine catabolism, and ROS metabolism (Table 1). With respect to ROS metabolism, ecotoxicological studies have revealed an impact of biopesticides on antioxidant defence in zebrafish, indicating oxidative stress and a decrease in peroxisomal catalase activity [41].

Orthologues of the 14 human peroxins have been identified, including Pex1/6/26, Pex2/10/12, Pex13/14, and Pex5/7, which are required for matrix protein import; Pex3/16/19, responsible for peroxisomal membrane protein sorting; and members of the Pex11 family, which regulates peroxisome proliferation (Figure 2). The *D. rerio* PTS1 receptor Pex5 contains the characteristic tetratricopeptide repeats at the C-terminus required for the interaction with the PTS1 cargo and a disordered N-terminal region, which harbours a Pex7-binding domain. Similar to humans and other eukaryotes, Pex5 (Pex5L) in zebrafish functions as a co-receptor for Pex7-mediated PTS2 import.

Furthermore, *D. rerio* has candidate genes encoding all the enzymes necessary for both peroxisomal and mitochondrial fatty acid beta-oxidation pathways with substrate spectra similar to those in humans (see Section 1).

Differences in peroxisomal metabolic pathways between humans and zebrafish have been revealed in bile acid synthesis and purine catabolism [21]. In humans, bile acid-CoA:amino acid N-acyltransferase (BAAT) catalyses the conversion of choloyl-CoA and deoxycholoyl-CoA to taurine- or glycine-conjugated cholic acid, or deoxycholic acid. Zebrafish and other fish species lack an orthologue of BAAT. This is likely due to the lack of C24 bile acids, this is likely due to the lack of C24 bile acids in *D. rerio* and other Cypriniformes, which are formed by side chain shortening in human peroxisomes. Thus, peroxisomes are not supposed to contribute to bile acid synthesis in *D. rerio* [42]. Similarly, ACOX2, a key enzyme in human bile acid biosynthesis, is also absent in zebrafish (Figure 1; Table 1). 

Peroxisomes contain enzymes involved in purine catabolism. However, there are differences in the expression and cellular localisation of those enzymes in vertebrate species. Fish and amphibians (and many invertebrates) express the purine-degrading enzymes xanthine oxidase, urate oxidase, allantoinase, and allantoicase, which generate the metabolites uric acid, allantoin, allantoic acid, and urea as well as ureidoglycolate (reviewed in [43]). In *D. rerio* and other freshwater fish, xanthine oxidase and allantoinase are cytosolic enzymes, whereas urate oxidase and allantoicase contain a PTS1, which targets them to peroxisomes [21,44]. Unlike zebrafish, most mammals excrete allantoin and therefore lack allantoinase and allantoicase. Moreover, humans (like other primates, birds, and reptiles) do not express functional uricase genes, and excrete uric acid [43,45] (Table 1). Similar to zebrafish, human xanthine oxidase is also cytosolic. In contrast to humans, *D. rerio* contains additional enzymes involved in urate degradation, namely 5-hydroxyisourate hydrolase (Uraha) and 2-oxo-4-hydroxy-4-carboxy-5-ureidoimidazoline decarboxylase (Urad), which may localise to peroxisomes [21]. These genes as well as the uricase gene were inactivated through pseudogenisation during hominoid evolution. 

Freshwater fish, including *D. rerio*, have an increased capacity to synthesise DHA compared to marine fish and mammals [46]. The omega-3 PUFA DHA (C22:6n-3) can be obtained through one round of peroxisomal beta-oxidation of C24:6n-3, which removes two carbons. In a recent study, Yang et al. [47] revealed a role for peroxisomal Enoyl-CoA hydratase/3-hydroxyacyl CoA dehydrogenase (*Ehhadh*) (L-BP in humans) in the synthesis of DHA in zebrafish. Two bifunctional enzymes (L-BP/MFP-1; D-BP/MFP-2) catalyse the 2nd and 3rd steps in peroxisomal fatty acid beta-oxidation, namely the hydration and dehydrogenation reactions (Figure 1). In mammals, L-BP is involved in the metabolism of dicarboxylic acids [48,49,50]. A dominant negative mutation in the EHHADH gene was identified in patients with an inherited form of renal Fanconi’s syndrome [51]. However, the pathophysiological cause was a disruption of mitochondrial oxidative phosphorylation caused by the mistargeting of the mutant EHHADH protein to mitochondria. Recent studies in an EHHADH deficient mouse model revealed male-specific kidney hypertrophy and proximal tubular injury [52].

EHHADH’s role in DHA production has been controversial, and a more prominent role for the D-bifunctional peroxisomal enzyme 17β-hydroxysteroid dehydrogenase type IV (D-BP/MFP-2; encoded by the HSD17B4 gene) has been shown in knockout mice [53,54]. The authors generated *Ehhadh*-deletion zebrafish (*Ehhadh*−/−) models using CRISPR/Cas9 technology [47]. *Ehhadh* deletion did not affect zebrafish survival and growth, but reduced the content of DHA and inhibited the synthesis of n-3 PUFA. Furthermore, an *Ehhadh* transgenic zebrafish model (*Tg:Ehhadh*) was generated, and overexpression of *Ehhadh* significantly increased the DHA content in the liver and upregulated the expression of genes related to PUFA synthesis [47]. The study of those zebrafish models revealed new insights into the role of *Ehhadh* in DHA synthesis in zebrafish. 

In contrast to the loss of Lbp, knockdown of Dbp in zebrafish caused severe abnormalities, including defective craniofacial morphogenesis, growth retardation, and abnormal neuronal development closely resembling the effects of D-BP mutations observed in humans [55] (see Section 4.3.2).

Overall, the comparison of the peroxisomal inventory in *D. rerio* and *H. sapiens* showed minimal differences in peroxisomal functions between the two species. This confirms the reliability of zebrafish as a vertebrate model for studying peroxisome biology.

## 4. Zebrafish Models of Peroxisomal Disorders

### 4.1. Peroxisome Biogenesis Disorders/Zellweger Spectrum Disorders

Zellweger Spectrum Disorders (OMIM #214100) represent a group of heterogeneous autosomal recessive disorders characterised by a defect in peroxisome biogenesis due to mutations in at least one of 14 *PEX* genes involved in the assembly of peroxisomes (Figure 2). The Zellweger spectrum is a clinical and biochemical continuum, with Zellweger syndrome (ZS) representing the most severe condition [56]. Patients can show severe symptoms in the neonatal period or present with milder symptoms later in life, during adolescence or adulthood. Due to a defect in peroxisome formation, multiple metabolic pathways are affected, usually resulting in the accumulation of VLCFAs, phytanic and pristanic acid, C27-bile acid intermediates in plasma, and a reduction in ether phospholipids/plasmalogens. Common clinical features are neurological abnormalities, loss of muscle tone (hypotonia), hearing and vision impairment, developmental delay, liver dysfunction, and kidney abnormalities. There is currently no curative therapy, but supportive care is available.

#### 4.1.1. *pex2* Mutant Zebrafish

Human PEX2 is one of the RING-containing peroxisomal membrane proteins, alongside PEX10 and PEX12 [57,58]. These proteins are crucial for ubiquitination processes required for receptor recycling [57,58,59] (Figure 2). A loss of PEX2 disrupts matrix protein import, leading to the formation of “ghost peroxisomes” devoid of matrix proteins and metabolic activities [60,61]. In humans, a PEX2 deficiency causes a severe ZS phenotype characterised by hypotonia, neurological abnormalities, and the accumulation of VLCFAs, and typically results in death within the first year of life [62].

The first zebrafish *pex2* deficient model was established by microinjecting TALENs (transcription activator-like effector nucleases) into zebrafish embryos at the 1-cell stage to induce a targeted double-strand break in the zebrafish *pex2* gene [63] (Table 2). Zebrafish Pex2 shares a 57% amino acid sequence identity with human PEX2 [63]. Mutant zebrafish showed phenotypes similar to human patients, including hepatic lipid droplet accumulation, reduced locomotion, feeding difficulties, and early death. Gene expression analysis revealed significant downregulation of muscle-related genes, such as troponin and parvalbumin, which are associated with calcium-binding functions [63]. This suggests that hypotonia in ZS may in part arise from muscular functional defects due to downregulation of these genes.

Elevated ER stress was also detected in the mutant fish, which could contribute to liver steatosis through the activation of stress pathways, as observed in *pex2* knockout mice [63,64,65,66,67]. This finding highlights a conserved link between peroxisomal deficiency, ER stress, and lipidosis in vertebrates [63]. Additionally, organ-specific VLCFA alterations were observed, including the accumulation of highly polyunsaturated VLCFAs in the brain and eyes, as well as a reduced expression of crystallin genes in the lens [63]. These changes might explain cataract formation in ZS patients [63,68].

A key advantage of the zebrafish *pex2* (and other *pex*) model is its ability to survive to adulthood, allowing the study of adult-specific ZS phenotypes, such as fertility and gametogenesis defects, which cannot be examined in ZS mouse models due to their early lethality [63]. Infertility was observed in the zebrafish, with females exhibiting underdeveloped oocytes and males producing normal motile sperm, suggesting sex-specific reproductive defects. These features establish the *pex2* mutant zebrafish as a valuable tool for investigating the metabolic and developmental aspects of ZS.

**Table 2 cells-14-00147-t002:** Zebrafish models for peroxisomal disorders.

Human Disorder	Targeted Gene	Method	Studied Tissues	Peroxisomal Phenotypes	Phenotypes Related to Clinical Features	Ref.
ZSD	*pex2*	TALENs- mediated knockout	Liver Brain Eyes Muscles	↓ Peroxisomal matrix protein import ↓ Catalase and glutathione peroxidase activities ↑ VLCFAs ↓ Ether phospholipids	△ Motor activity Hypotonia ↑ Hepatic lipid △ Neuronal and muscle function △ Gametogenesis ↓ Crystallin genes ↓ Survival	[63]
ZSD	*pex5*	CRISPR/Cas9-mediated knockout	Liver Nervous system Whole larvae	↓ Peroxisomal matrix protein import ↓ Peroxisome abundance	△ Motor activity Demyelination ↑ Hepatic lipid Edema Deflated swim bladder Shrunken liver ↓ Survival Expedited death under fasting conditions	[69]
ZSD	*pex13*	CRISPR/Cas9-mediated knockout	Liver Whole larvae	↓ Peroxisomal matrix protein import ↓ Peroxisome abundance ↑ Ubiquitinated PEX5 ↑ Peroxisome-dependent ROS ↑ Pexophagy	△ Motor activity ↑ Hepatic lipid accumulation △ Neuronal function Liver steatosis Laval mortality	[38]
	*pex13*	Morpholino- mediated knockout (mosaic)	Liver Pronephric duct Yolk sac	Partial elimination of peroxisomes	Not assessed	[30]
ALD	*abcd1*	TALENs- mediated knockout	CNS Adrenal glands Whole embryo	↑ VLCFAs ↑ Cholesterol	△ Motor activity Hypomyelination △ Oligodendrocyte patterning △ CNS development ↑ Apoptosis ↓ Survival	[70]
Mitchell Syndrome	*acox1*	Transient overexpression (N237S)	Brain Spinal cord Whole embryo	↓ Peroxisome density ↑ Oxidative stress	△ Motor activity Activation of ISR ↓ Survival	[71]
D-BPD	*dbp*	Morpholino- mediated knockdown	Whole embryo Liver Pancreas	↓ beta-oxidation ↓ Ether phospholipids synthesis ↓ *pex5* expression	△ Yolk lipid consumption Growth retardation Morphological malformation △ Neuronal, liver, pancreas, cartilage, blood, blood vessels, digestive organ development Abnormal vascular patterning Embryonic lethality	[55]
Osteoarthritis	*fis1*	Morpholino- mediated knockdown	Whole embryo	↓ Peroxisome abundance ↓ Catalase and glutathione peroxidase activities ↓ beta-oxidation gene expression	↑ Apoptosis ↑ Lipid ↓ Survival	[37]
Unspecified	*vwa8*	Morpholino- mediated knockdown	Whole embryo	Not assessed	△ Motor activity Developmental delays and defects Light sensitivity Facial dysmorphism ↓ Survival	[72]
Autosomal-dominant retinitis pigmentosa	*vwa8*	Morpholino- mediated knockdown	Retina Photoreceptor layer	Not assessed	△ Visual function Retinal pigment deposition Thinning of the retinal photoreceptor layer	[73]

↑, Upregulation; ↓, Downregulation; △, Impairment; ZSD, Zellweger Spectrum Disorder; ALD, adrenoleukodystrophy; D-BPD, D-bifunctional protein deficiency; VLCFAs, Very Long Chain Fatty Acids; ROS, Reactive Oxygen Species; CNS, Central Nervous System; ISR, integrated stress response.

#### 4.1.2. *pex3* Mutant Zebrafish

PEX3 is essential for peroxisome membrane assembly via class I and class II pathways [58,74] (Figure 2). In the class I pathway, PEX19 forms complexes in the cytosol with newly synthesised peroxisomal membrane proteins (PMPs) including PEX16 and carries them to the receptor PEX3 [58]. The PEX3 on the peroxisomal membrane serves as an anchoring site for PEX19-PMP complexes, facilitating membrane assembly [58]. In the class II pathway, PEX16 acts as a receptor for PEX19-PEX3 complexes, mediating the transport of newly synthesised PEX3 to the peroxisome [58]. As PEX3 is critical for forming peroxisomal membrane structures, its deficiency results in the loss of peroxisomes [75].

Early research on *pex3* in zebrafish was conducted by overexpressing the N-terminal 45 amino acids of human PEX3 (PEX3_(1–45)_–GFP), which was previously shown to induce peroxisome degradation by pexophagy in human skin fibroblasts [76]. However, overexpression of this dominant negative PEX3_(1–45)_–GFP in zebrafish did not affect peroxisomal import [30]. Moreover, PEX3_(1–45)_–GFP overexpressing fish exhibited comparable peroxisomal catalase activity to wild type fish, suggesting that dominant negative overexpression was not efficient to inactivate *pex3* or eliminate peroxisomes [30]. Alternative molecular approaches, such as CRISPR/Cas9-mediated knockout, are required to establish a definitive *pex3*-deficient zebrafish model.

#### 4.1.3. *pex5* Mutant Zebrafish

PEX5 is essential for importing peroxisomal matrix proteins containing the C-terminal peroxisomal targeting signal PTS1 as well as for supporting PEX7 to import cargos containing the N-terminal PTS2 signal [58] (Figure 2). Thus, a genetic disruption of PEX5 results in the loss of peroxisomal matrix protein import and the formation of ghost peroxisomes [62,77,78].

Zebrafish Pex5 shares a 71.3% amino acid sequence similarity with human PEX5 [69]. Homozygous *pex5* knockout zebrafish (*pex5*^−/−^), generated using CRISPR/Cas9, recapitulate key features of ZS, including loss of functional peroxisomes, lipid accumulation, demyelination, and early death within one month after birth (healthy zebrafish live for around three years) [69] (Table 2). Given that dysfunctional peroxisomes can induce mitochondrial abnormalities [79,80,81,82], fasting experiments were conducted in *pex5*^−/−^ zebrafish to evaluate the effects of *pex5* loss and nutritional stress on their mitochondria [69]. Fasting exacerbated mortality in *pex5*^−/−^ due to deregulated mitochondrial function and impaired mTORC1 signalling, which led to nutrient depletion and mitochondrial damage in the liver [69]. Therapeutic interventions targeting mitochondrial function, mTORC1 signalling, oxidative stress, or autophagy activation ameliorated metabolic imbalances and extended survival in fasted *pex5*^−/−^ zebrafish [69]. These findings in *pex5*^−/−^ zebrafish highlight fasting as detrimental when peroxisomes are dysfunctional and suggest potential therapeutic strategies for ZS-related metabolic disorders [69].

#### 4.1.4. *pex13* Mutant Zebrafish

PEX13 is a peroxisomal membrane protein characterised by an Src homology 3 (SH3) domain located at its C-terminus, which faces the cytosol [83,84,85]. PEX13 transiently interacts with PEX14, another peroxisomal membrane protein, to facilitate peroxisomal matrix protein import [86,87,88] (Figure 2). The SH3 domain of PEX13 binds to PEX5, enabling the transport of PTS1- and PTS2-containing cargo from the cytosol into peroxisomes [38,89,90]. The loss of PEX13 leads to a marked reduction in PEX5 associated with the peroxisomal membrane and disrupts the import of both PTS1 and PTS2 proteins [84,90]. Therefore, patients with PEX13 deficiency have defects in the import of peroxisome matrix proteins, resulting in ghost peroxisomes as well as the accumulation of VLCFAs [91,92,93,94]. A Pex13 knockout mouse model also shows several clinical features of ZS, such as a loss of peroxisome function, severe hypotonia, failure to feed, and neonatal death [90].

The first *pex13*-deficient zebrafish model was generated using morpholinos to block *pex13* translation [30] (Table 2). Although this approach partially reduced the number of peroxisomes in the liver, pronephric duct, and wall of the yolk sac, it proved insufficient due to dilution and uneven distribution of the morpholinos during embryonic development [30].

A more recent study employed a CRISPR-Cas9 approach to establish homozygous *pex13* knockout zebrafish (*pex13*^−/−^) [38] (Table 2). Interestingly, while *pex13*^−/−^ zebrafish obtained from heterozygous knockout in-crosses survived into adulthood, maternal and zygotic *pex13* KO (*MZpex13*^−/−^) zebrafish generated from *pex13*^−/−^ parents died during the larval stage. This indicates a crucial role of Pex13 during early zebrafish development [38]. *MZpex13*^−/−^ zebrafish showed a reduced number of peroxisomes and defects in matrix protein import. Additionally, the mutant zebrafish exhibited a high mortality rate and a dark lipid phenotype due to lipid accumulation in the liver [38]. The study also uncovered a novel role of PEX13 in regulating pexophagy. The loss of PEX13 enhanced pexophagy through two mechanisms: (i) accumulating ubiquitinated PEX5 on the peroxisomal membrane, which recruits autophagy receptors, and (ii) increasing cellular ROS levels, thereby promoting autophagosome formation [38]. Supporting this, the reduction of peroxisome numbers in *MZpex13*^−/−^ fish was corrected by autophagy inhibitor treatment [38]. However, pexophagy inhibition did not reverse liver lipid accumulation in *MZpex13*^−/−^ fish, and instead increased hepatic lipid accumulation in both the WT and *pex13* knockout fish, suggesting that autophagy may be required for lipid droplet turnover in the liver [38].

### 4.2. Peroxisomal Single Enzyme Deficiency

#### Adrenoleukodystrophy (ALD) Model Zebrafish

Adrenoleukodystrophy (OMIM #300100) is one of the most common among leukodystrophies and peroxisomal disorders (incidence 1:14,000), caused by mutations in the X-linked *ABCD1* gene [95,96]. This gene encodes a peroxisomal ABC half-transporter for VLCFAs [96] (Figure 2). Dysfunctional ABCD1 leads to the accumulation of VLCFAs in plasma and tissues, as they cannot be imported into the peroxisome for degradation by beta-oxidation [96]. This results in neurological abnormalities and adrenal insufficiency. ALD shows phenotypic heterogeneity and can present as cerebral ALD, a severe demyelinating form affecting boys in their childhood, or as adrenomyeloneuropathy (AMN), a progressive peripheral myelopathy affecting adult males and females.

The amino acid sequence of zebrafish Abcd1 shows substantial homology to human ABCD1, with analogous expression patterns observed in the central nervous system and adrenal glands [70]. A zebrafish ALD model was established using TALENs to introduce a premature stop codon in *abcd1* [70] (Table 2). This mutant zebrafish model successfully mirrors key features of human ALD, including elevated VLCFA levels, cholesterol accumulation, impaired development of the central nervous system (CNS) and interrenal organ (representative of the mammalian adrenal cortex), and decreased lifespan [70,97]. Notably, CNS dysfunction in the model includes spinal cord hypomyelination, impaired motor functions, and altered oligodendrocyte patterning due to increased apoptosis. Motor phenotypes emerge as early as the first week post-fertilisation [28,70,95]. Furthermore, the zebrafish model revealed previously-unrecognised developmental defects in oligodendrocyte generation and patterning, emphasising the essential role of *abcd1* in oligodendrocytes, which may be critical for understanding ALD pathophysiology [70].

The early manifestation of the disease, with impaired motor/swimming functions of the *abcd1* mutants, enabled the development of a functional motor behaviour assay for high-throughput drug screening [28]. Remarkably, chloroquine, one of the top hits of the screen, rescued the motor behaviour of the *abcd1* mutant zebrafish. It increased the expression of stearoyl-CoA desaturase-1 (*scd1*), an orthologue of the human SCD1, which alleviated the lipid toxicity of accumulating saturated VLCFAs by fatty acid desaturation and metabolic re-routing of fatty acid synthesis towards the generation of less-toxic mono-unsaturated VLCFAs [28]. Conversely, *scd* and *scdb* knockout zebrafish, obtained using CRISPR/Cas9, also presented a motor deficit [28]. Chloroquine also successfully reduced saturated VLCFAs in ALD patient fibroblasts and increased SCD1 levels. Furthermore, treatment of *Abcd1*^-/y^ mice with agonists of the liver X receptor (LXR) increased SCD1 expression and reduced VLCFAs in relevant tissues. Metabolic re-routing of saturated to mono-unsaturated VLCFAs may thus represent a therapeutic strategy to alleviate VLCFA toxicity in ALD and other peroxisomal disorders with VLCFA accumulation [28].

Collectively, the zebrafish ALD model has been invaluable for high-throughput drug screening to identify therapeutic compounds for ALD patients and an improved understanding of the toxicity of saturated VLCFAs in ALD [28,95].

### 4.3. Peroxisomal Beta-Oxidation Deficiency

#### 4.3.1. ACOX1 Mutant Disease Model Zebrafish

ACOX1 (acyl-CoA oxidase 1) is an enzyme involved in the initial step of the peroxisomal fatty acid beta-oxidation pathway, where VLCFAs are broken down [71] (Figure 1).

Loss-of-function (LOF) mutations in *ACOX1* cause a rare and severe autosomal recessive disorder known as peroxisomal acyl-CoA oxidase deficiency or pseudoneonatal adrenoleukodystrophy (P-NALD) (OMIM #264470), characterised by VLCFA accumulation [98]. P-NALD patients manifest with infantile-onset hypotonia, leukodystrophy, seizures, visual and hearing impairments, loss of motor achievements, and progressive grey matter degeneration [95]. Although a zebrafish model for P-NALD has not been reported to date, an ACOX1 LOF mutant was generated using CRISPR/Cas9 methods in *Drosophila*, which highly and specifically express ACOX1 in glial cells in the central nervous system (CNS) and peripheral nervous system (PNS) [99]. This *Drosophila* model shows glial degeneration caused by VLCFA accumulation [99]. Additionally, ACOX1 LOF may also trigger an inflammatory response as the IL-1 inflammatory pathway is upregulated in ACOX1 mutant patient fibroblasts [98,100]. In *ACOX1*^−/−^ mice, however, these neuroimmune changes have not been described [100,101].

In contrast, gain-of-function (GOF) missense mutations in *ACOX1* (c.710A>G; p.N237S) result in Mitchell Syndrome (OMIM #618960), a rare neurodegenerative disorder characterised by sensorineural hearing loss, polyneuropathy, cognitive decline, and seizures [71,99,102]. Patients with the ACOX1 (p.N237S) mutation show significant losses of Schwann cells and neurons [99]. Studies using *Drosophila* and primary rat Schwann cell cultures have shown that the human ACOX1^N237S^ mutation promotes dimerisation and accumulation of ACOX1, leading to upregulated enzymatic activity, increased oxidative stress, and glial damage. These pathological processes result in the degeneration of glial cells and axons and drive the clinical features of Mitchell Syndrome [71,99]. Unlike ACOX1 LOF mutations, Mitchell Syndrome does not result in elevated VLCFA levels [71].

Zebrafish Acox1 shares a 70% amino acid sequence identity to human ACOX1, including conservation of the asparagine 237 residue, which is critical for Mitchell Syndrome onset [71]. The first vertebrate model of Mitchell Syndrome was established in zebrafish by transiently overexpressing the human ACOX1^N237S^ variant tagged with GFP, using the Tol2 transposon system under a β-actin promoter [71] (Table 2). This zebrafish model displayed reduced swimming ability, reflecting decreased motor activity, as well as activation of the integrated stress response (ISR) and reduced peroxisome density and number [71]. Notably, treatment with antioxidants rescued the swimming defects in ACOX1^N237S^ zebrafish. However, the model did not show any changes in oligodendrocyte numbers, whereas primary rat Schwann cells from the PNS showed a reduction in cell number upon ACOX1^N237S^ overexpression. This suggests differential susceptibility to the mutation between CNS oligodendrocytes and PNS Schwann cells [71,99].

#### 4.3.2. D-Bifunctional Protein (Dbp) Deficiency Model Zebrafish

D-bifunctional protein (D-BP), also known as 17-β-hydroxysteroid dehydrogenase type 4, as well as multifunctional protein 2 (MFP-2), is an enzyme responsible for the second and the third reactions of the four-step fatty acid beta-oxidation in peroxisomes [103,104] (Figure 1). Mutations in human D-BP lead to D-BP deficiency (D-BPD) (OMIM #261515), which causes severe neonatal abnormalities such as growth retardation, neuropathy, craniofacial malformation, and hypotonia within an early period of life [55,103,104,105,106,107,108]. The clinical presentation of P-NALD and D-BPD resembles that of the PBDs, even though there are defects in only a single enzyme [62]. D-BPD causes reduced oxidation of VLCFAs, pristanic acid, and di- and trihydroxycholestanoic acids (DHCA, THCA), resulting in their accumulation in the plasma of patients [62,104,108].

D-BP is well conserved across vertebrates and ubiquitously expressed in zebrafish [55]. Morpholino-based *dbp* knockdown in zebrafish embryos recapitulates symptoms seen in D-BPD patients, such as morphological malformations, defective yolk consumption, abnormal neuronal development, and growth retardation [55] (Table 2). Notably, *dbp* knockdown in zebrafish significantly reduced the expression of genes involved in peroxisomal functions, such as enzymes for ether phospholipid synthesis, and genes for mitochondrial biogenesis, as early as 1 day post-fertilisation (1 dpf) [55]. The zebrafish D-BPD model also revealed novel phenotypes not previously reported in humans or mice, such as impaired development of blood, blood vessels, and endoderm-derived organs (e.g., liver, pancreas) [55]. This distinction may arise from the transcriptional changes triggered by *dbp* knockdown at 1 dpf, a critical development stage when most organs have not yet formed. By contrast, DBP-deficient mouse models and human patients are examined at several weeks postnatally, when organs are already developed [53,54,55,104,105,106,107,109]. These unique features of the DBP zebrafish model provide an invaluable opportunity to investigate the early developmental symptoms and mechanisms of D-BPD.

### 4.4. Dually Targeted Peroxisomal/Mitochondrial Protein Deficiency

Peroxisomes and mitochondria cooperate in the metabolism of cellular lipids and ROS. Their functional interplay also includes cooperation in antiviral signalling and coordinated biogenesis by sharing key division proteins [14]. Several membrane and matrix proteins are therefore dually targeted to both peroxisomes and mitochondria [110]. These include the tail-anchored membrane proteins FIS1 and MFF, which contribute to organelle division and multiplication (Figure 2).

#### 4.4.1. Fission 1 (FIS1) Deficiency Model Zebrafish

FIS1 is a tail-anchored adaptor protein dually targeted to peroxisomes and mitochondria [111]. On the mitochondrial surface, FIS1 collaborates with another tail-anchored protein, MFF (mitochondria fission factor), to recruit division machinery such as the GTPase DRP1 [111,112,113]. On the peroxisome surface, FIS1 works not only with MFF but also with PEX11β, a peroxisomal membrane protein, to drive peroxisome division [111]. Uniquely, FIS1 and PEX11β can also independently promote peroxisome division in the absence of MFF, highlighting their distinctive roles in peroxisomal dynamics [111].

While no patients with a mutation in the *fis1* gene have yet been reported, FIS1 is significantly suppressed in human osteoarthritis (OA) chondrocytes [37]. Fis1 was knocked down in zebrafish using morpholinos to specifically block either translation or splicing [37] (Table 2). Morpholino-injected zebrafish embryos showed reduced peroxisome abundance, decreased catalase and glutathione peroxidase activities, abnormal lipid accumulation, increased cell death, and abnormal development [37]. Additionally, *fis1*-deficient zebrafish displayed lysosomal accumulation and mitochondrial dysfunctions [37]. In human OA chondrocytes, *fis1* suppression led to lysosomal accumulation and inhibition, altered miRNAs expression (particularly those involved in lysosomal regulation), chondrocyte apoptosis, and suppression of autophagy due to lysosomal destruction [37]. These findings suggest that *fis1* dysregulation impairs peroxisomal, mitochondrial, and even lysosomal functions. Crucially, lysosomal impairment ultimately stimulates apoptosis through the suppression of pexophagy, highlighting a critical role for FIS1 in cellular homeostasis through regulating organellar functions [37].

#### 4.4.2. VWA8 Deficiency Model Zebrafish

P7BP2/VWA8 (Pex7p-binding protein 2/von Willebrand factor domain-containing 8) (also named KIAA0564) was identified by Niwa et al. [114] as a new PEX7 binding protein. Indeed, the human protein possesses two predicted PTS2 sequences in the N-terminal domain (aa.66–74 and aa.71–79), and deletion of these regions resulted in the loss of peroxisome targeting and PEX7 binding. P7BP2/VWA8 further possesses six putative AAA+ (ATPases associated with diverse cellular activities) domains and a putative von Willebrand factor A domain at the C-terminus. It has been suggested that P7BP2/VWA8 represents a novel dynein-type AAA+ family protein as, similar to dynein, the AAA+ domains are arranged in a hexameric ring structure [114]. A peroxisomal localisation of P7BP2/VWA8 is also supported by proteomics studies of mouse kidneys [115]. Interestingly, P7BP2/VWA8 belongs to a group of dually targeted proteins which localise to peroxisomes and mitochondria. Luo et al. [116] revealed that the mouse P7BP2/VWA8 contains an N-terminal mitochondrial targeting signal (aa.1–34) and localises to mitochondria in mouse AML12 cells, whereas deletion of these amino acids resulted in cytosolic localisation. They also reported that the protein localises to the matrix side of the inner mitochondrial membrane [117]. P7BP2/VWA8 knockout in AML12 mouse hepatocytes resulted in pathological conditions including oxidative stress, protein degradation, upregulation of HNF4a, and increased expression of mitochondrial (higher complex 1, ATP synthase levels) and peroxisomal proteins along with lipid transport proteins [118,119]. Furthermore, increased mitochondrial compensatory and oxidative capacity in the knockout cells was observed [119], suggesting a link to energy metabolism. It has been proposed that 7PBP2/VWA8 may play an important role in protein quality control in peroxisomes and mitochondria [120], possibly as an AAA+ unfoldase. A putative role in unfolding porphyrin or corrin rings to insert chelated magnesium or another cationic metal in mitochondria has been suggested [121]. 

Recently, patients with a defect in VWA8 (OMIM #617509) have been identified, exhibiting severe developmental disorders. Symptoms include global developmental delay, spastic diplegia, microcephaly, scoliosis, pneumonia, dyspnoea, fever, progressive inability to walk, cardiovascular anomalies, brain atrophy, Achilles tendon contracture, lower limb hypertonia, limb hypertonia, hyperactive deep tendon reflexes, thoracic scoliosis, abnormality of the hip bone, and abnormality of the sphenoid sinus [72]. A homozygous missense variant [c.947A>G; p.(Asp316Gly)] was identified in exon 8 of the *VWA8* gene, potentially destabilising the VWA8 protein structure and thus disrupting VWA8 function.

To investigate the developmental impact of VWA8, zebrafish *vwa8* knockdown was performed using morpholinos (Table 2). Zebrafish *vwa8*, which is located on chromosome 9 (Gene bank accession number NM_001128338.1), shares a high similarity and conserved domains with human VWA8, including a predicted mitochondrial targeting sequence [72]. Morpholino-treated zebrafish (*vwa8* morphants) displayed various phenotypes, including developmental delay at an early stage, cardiovascular anomalies (such as cardiac edema and cardiac hypertrophy), a disorganised notochord, severe skeletal anomalies (e.g., scoliosis), impaired locomotion, light sensitivity, and facial dysmorphism. The expression of *vwa8* mRNA in oocytes indicates that it is a maternally transcribed gene with important roles in early embryonic development in zebrafish. Overall, these findings support an association of VWA8 with complex neurodevelopmental and skeletal disorders.

Moreover, heterozygous mutations in *VWA8* [missense variant c.3070G>A (p.Gly1024Arg) and nonsense variant c.4558C>T (p.Arg1520Ter)] have been linked to autosomal-dominant retinitis pigmentosa (OMIM #268000), the most common type of hereditary retinal dystrophy [73]. VWA8 expression in those individuals was reduced as a result of the premature termination of translation. VWA8 was found to be specifically expressed in the retina of eye tissues, suggesting that VWA8 plays an important role in retinal development. The exact mechanism of the pathogenesis of retinitis pigmentosa is unclear, but it is accompanied by oxidative damage in photoreceptors and retinal pigment epithelial cells. 

To determine whether the loss of VWA8 leads to retinitis pigmentosa, a knockdown zebrafish model was developed (Table 2). Knocking down of VWA8 induced abnormal phenotypes, including retinal pigment deposition, eyes with damaged photoreceptor cells, and severe malformation, with significantly higher fetal mortality and morbidity compared to controls [73]. Additionally, a significant narrowing of the retinal photoreceptor layer was observed in *Vwa8*-knockdown larvae (5 dpf) as well as aberrant locomotor response during the light-off phase/dark adaptation. These findings confirm an important role of Vwa8 in early embryo development and in early retinal development. 

An siRNA-mediated VWA8 knockdown in ARPE-19 cells (spontaneously arising retinal pigment epithelia cell line) revealed mitochondrial damage, with decreased mitochondrial membrane potential, an increase in mitochondrial ROS production, and decreased respiration and ATP production. Furthermore, the activation of mitophagy and apoptosis was observed, resulting in cell degeneration, which could explain retinal damage. 

Overall, these studies suggest that VWA8 is important for mitochondrial function. However, its specific role and the molecular mechanism of VWA8 deficiency on mitochondrial function requires further studies. This also applies to its role in peroxisomes. As peroxisomes also contribute to cellular redox homeostasis, and intimately cooperate with mitochondria, e.g., in fatty acid beta-oxidation and energy homeostasis, they may contribute to the observed pathophysiology. Importantly, peroxisomal dysfunction has been linked to developmental processes, neurodegeneration, and retinal damage.

## 5. Conclusions

The ability to investigate developmental and neurological alterations as well as lipid metabolism in zebrafish has made them an attractive vertebrate model for the study of peroxisomal disorders. They show a high degree of genome conservation with mammals; this is also reflected in their peroxisomal gene and protein inventory. It should be noted that, due to a genome-wide gene duplication during evolution, zebrafish often harbour two copies of genes, including some peroxisomal genes.

The comparison of the *D. rerio* peroxisomal inventory with *H. sapiens* revealed the presence of the major metabolic pathways of peroxisomes as well as peroxisomal proteins. The fatty acid substrate profiles metabolised in peroxisomes (or mitochondria) closely mirror those observed in humans. In addition, the peroxisomal protein targeting signals are conserved. Variations in peroxisomal metabolism have only been observed for bile acid synthesis and purine catabolism.

The current zebrafish models for peroxisome biogenesis disorders/Zellweger spectrum disorders (e.g., Pex2, Pex5 deficiency) show phenotypes similar to human ZSD patients (e.g., locomotor impairments, feeding difficulties, liver dysfunction, demyelination, early mortality) as well as alterations in fatty acid metabolism (e.g., accumulation of VLCFAs, branched chain fatty acids, reduced ether phospholipid synthesis). Furthermore, they revealed previously unrecognised defects. An advantage of the zebrafish Pex deficiency models is their ability to survive to adulthood. This allows investigation of adult-specific ZS phenotypes (e.g., fertility, gametogenesis defects), which are often difficult to study in ZS mouse models due to early lethality. They may therefore become useful models for the study of sex-specific differences due to peroxisomal dysfunctions.

Many of the peroxisomal disorders that affect the CNS and zebrafish models also capitulate the features. The blood-brain barrier (BBB) in zebrafish shares several key similarities with humans, including the presence of endothelial cells with tight and adherent junctions, reduced transcytosis, and a neurovascular unit (NVU) comprising endothelial cells, pericytes, and glial cells [122]. Humans and zebrafish also exhibit conserved transport mechanisms, such as glucose transport via Glut1/Slc2a1, and rely on similar signalling pathways like vascular endothelial growth factor (VEGF) and Wnt/β-catenin for BBB development. However, zebrafish lack the stellate astrocytes found in humans, relying instead on radial glia, which perform some analogous functions [122]. Zebrafish also show differences in the timeline of BBB maturation, which occurs earlier (around 2.5–3 dpf), and in transporter genes, lacking the human ABCB1 efflux transporter but expressing homologues like ABCB4 and ABCB5. Despite these differences, zebrafish remain a valuable model for studying BBB development and function, with findings broadly translatable to humans. This is also important with respect to brain-related drug screening for peroxisomal disorders.

The *abcd1* mutant zebrafish model exhibits features of human ALD (e.g., elevated VLCFA levels, alterations in CNS and the interrenal organ) and has been successfully used for high-throughput drug screening to identify therapeutic compounds for ALD patients. This highlights another advantage, as screening assays can be developed using the restoration of motor behaviour/swimming functions as a read-out. In this respect, zebrafish may also become useful models for the study of Pex mutations with a milder phenotype or other single enzyme deficiencies. In the future, we therefore anticipate zebrafish models to become a fundamental resource for the peroxisome community, not only to study peroxisomal biology, but also to understand and ultimately develop therapies for a range of peroxisomal diseases.

## Figures and Tables

**Figure 1 cells-14-00147-f001:**
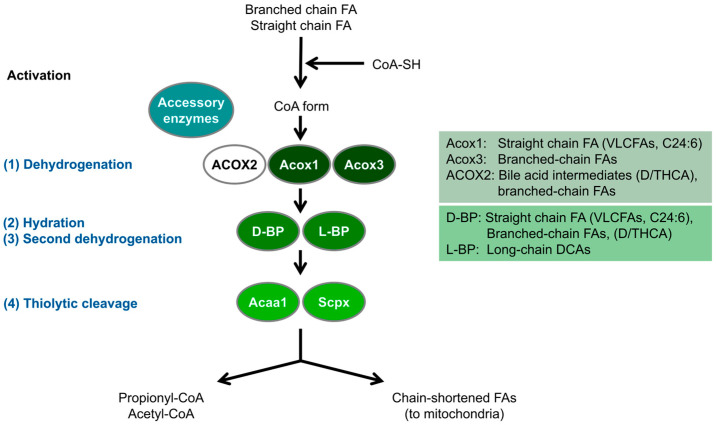
Schematic overview of the peroxisomal fatty acid beta-oxidation pathway in *D. rerio*. Peroxisomes metabolise fatty acids (FA) through a four-step beta-oxidation process; **1. Dehydrogenation**: Catalysed by multiple acyl-CoA oxidases with different substrate specificities. In humans, ACOX2 specifically oxidises C27 bile acid intermediates (D/THCA), but this enzyme is absent in zebrafish (see also Table 1). **2. Hydration**: Enzymatic activity is mediated by two bifunctional proteins (L-BP, D-BP) with enoyl-CoA hydratase and 3-hydroxy-acyl-CoA dehydrogenase functions. **3. Second dehydrogenation**: Also mediated by the bifunctional proteins. **4. Thiolytic cleavage**: Carried out by various thiolases, including Acca1 and Scpx. Additionally, peroxisomes utilise accessory enzymes, such as those involved in fatty acid alpha-oxidation, to modify the acyl-CoA esters of those fatty acids that cannot directly enter the beta-oxidation pathway (e.g., phytanic acid) (see text for details) (adapted from [18]). These enzymes ensure the effective metabolism of diverse fatty acids. Acaa1, acetyl-Coenzyme A acyltransferase 1; DCA, dicarboxylic acid; D/THCA, di-/tri-hydroxycholestanoic acid; Scpx, sterol carrier protein x.

**Table 1 cells-14-00147-t001:** Overview of *D. rerio* peroxisomal metabolic pathways in comparison to *H. sapiens*.

Function/Pathway	Comments	*Dr*	*Hs*
Fatty acid beta-oxidation	Acox2 is absent in zebrafish ACOXL/ACOX4 is present in humans and zebrafish, but not well defined		
Bile acid synthesis	Bile acid-CoA:amino acid N-acyltransferase (Baat) is absent in zebrafish	-	
Fatty acid alpha-oxidation			
Saturation of PUFAs			
Ether lipid synthesis			
Glycolate/ Glyoxylate metabolism	Hydroxyacid oxidase 3 (Hao3) is absent in zebrafish		
Amino acid catabolism	D-amino acid oxidases 2 and 3 (Dao2, Dao3) are absent in humans		
Amine metabolism			
Purine and pyrimidine metabolism	Urate oxidase (Uricase) (Uox), Allantoicase (Allc), Urate (5-hydroxyiso-) hydrolase a (Uraha), Ureidoimidazoline decarboxylase (Urad) are absent or inactivated in humans		-
Oxygen metabolism/ Oxidation redox equivalents			
Proteases			
Carbohydrate metabolism			

*Dr*, *D. rerio*; *Hs*, *H. sapiens*. □—absent/not identified; ■—pathway present, peroxisomal proteins identified; ■—pathway/proteins present, with some alterations (see comments) (adapted from [18]).

## Data Availability

All datasets generated for this study are included in the article.

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
