# Peer review of "Modelling Peroxisomal Disorders in Zebrafish"

_cells, 2025, doi:10.3390/cells14020147_

Round 1
Reviewer 1 Report
Comments and Suggestions for Authors
The review entitled “Modelling Peroxisomal Disorders in Zebrafish” by Jiang and Schrader is a well-written and comprehensive summary of the knowledge regarding the role of peroxisomes in zebrafish, as well as a comparison to humans. The review could already be published as is. I only have a few minor suggestions that, depending on perspective, might improve the review.
These are just suggestions:
1. In line 40 next to ref 8 it would be nice it also PMID: 36085307 could be cited at it has revealed a very general antiviral response involving peroxisomes.
2. Line 63 since the reference 11 (still important for the first mentioning) enzymes have been discovers and more recent reviews summarizing the biochemical pathway up to date PMID: 36120579
3. With regard to Fig 2: it is a very nice figure – I like it very much. Would be nice to add Far1 on the other side and name it “Fatty alcohol for ether lipid synthesis”. If it is placed in the corner between Acbd4Vap tether and Mavs it might fit. Just a suggestion for completeness and more of similarity to humans. You might also change the ration of PTS1 and PTS2 proteins as the majority of the peroxisomal matrix proteins of D. rerio possess a PTS1, and only a few contain a PTS2.
4. Line 355 (Pex5): it would be nice to indicate the normal live expectation hear to allow the reader to understand the meaning of the death with one month after birth.
5. Table 2: What is aka as human disorder for pex5 deficiency shouldn’t this be also ZSD?
6. Lain 448 (also 489): why not going with: OMIM #264470 PEROXISOMAL ACYL-CoA OXIDASE DEFICIENCY instead of P-NALD? This review otherwise went with the OMIM nomenclature. There is also a number for the Mitchell Syndrome: OMIM #618960. In general the OMIM # could be added for all disorders when the first time mentioned.
7. At lain 645 (or on any other place): As most of the peroxisomal disorders affect the central nervous system a brief discussion on the similarities and differences between the blood-brain barrier (BBB) in humans and zebrafish could offer valuable insights, as understanding these variations is crucial for assessing the relevance of zebrafish models in brain-related drug discovery. This seems of particular interest in the context zebrafish in drug screening to identify novel medications, particularly with the growing availability of drug libraries that can be efficiently tested through high-throughput methods.
Reviewer 2 Report
Comments and Suggestions for Authors
The manuscript summarized and compared the metabolism pathways and peroxin proteins between zebrafish and mice and described peroxisome function deficient zebrafish models. The review is comprehensive and helpful for zebrafish and peroxisome field. I have some minor questions as indicated below.
1. Line 504: “By contrast, DBP-deficient mice models and human patients are examined at 504 several weeks postnatally, when organs are already developed [51–53,100–103,105]”
This explanation is not clear. Are the DBP-deficient mice inducible model where the knockout is induced several weeks postnatally? But how about the DBP human patients? Are the zebrafish-specific development defects transient and recovered in the adult stages?
2. Line 536: These findings suggest that fis1 dysregulation impairs peroxisomal, mitochondrial and lysosomal functions by altering miRNAs expression.
The mechanisms of how fis1 affects miRNA expression to impair peroxisomal, mitochondrial and lysosomal functions is lacking in the manuscript. The statement is not convincing.
3. Line 562: Based on those data, 7PBP2/VWA8 may play an important role in protein quality control in both peroxi- 562 somes and mitochondria [116], possibly as an AAA+ unfoldase.
This conclusion is not supported by the data described. The change of protein levels due to p7bp2/vwa8 knockout is not sufficient to indicate that this gene is involved in protein quality control. Please provide more data to support it.
4. Line 541: The manuscript did not provide any data showing VWA8 deficiency leading to peroxisome defects. Please include them if there is any.
5. The authors claimed that one advantage of peroxisome deficient zebrafish model is that these fish can survive to adulthood while the mouse model could not. Can authors explain why loss of peroxisome function are not as lethal in zebrafish as in mice.
